# A Genome-Wide Analysis of the WUSCHEL-Related Homeobox Transcription Factor Family Reveals Its Differential Expression Patterns, Response to Drought Stress, and Localization in Sweet Cherry (*Prunus avium* L.)

Fei Deng [1,2], Hongming Wang [1,2,*], Xiaojuan An [1,2] and Jean Yves Uwamungu [1,3]

[1]   College of Bioengineering and Biotechnology, Tianshui Normal University, Tianshui 741000, China; dengfei@tsnu.edu.cn (F.D.); anxj0301@163.com (X.A.); joady9@yahoo.fr (J.Y.U.)
[2]   Sweet Cherry Technology Innovation Center of Gansu Province, Tianshui 741000, China
[3]   Gansu Provincial Key Laboratory for Utilization of Agricultural Solid Waste Resources, Tianshui 741000, China
[*]   Correspondence: wanghm226@tsnu.edu.cn

**Abstract:** The WUSCHEL-related homeobox (*WOX*) gene family has a critical effect on plant development and abiotic stress. However, there have been no genome-wide studies on *WOX* genes within sweet cherry (*Prunus avium* L.). In the present work, eight *PavWOX* genes were discovered within sweet cherry at the genome-wide level, and they were mapped to six chromosomes. Based on phylogenetic relationships, these genes were classified into three groups, with genes in one group having similar gene structures and conserved motifs. Meanwhile, the *PavWOX* genes possessed cis-acting elements and functions associated with hormone responses, stress responses, and development. As revealed by expression patterns, certain *PavWOX* genes are specifically expressed within tissues, suggesting that they may have unique functions. Additionally, the gene family expression patterns under drought stress were analyzed. *PavWOX4*, *PavWOX5*, *PavWOX13A*, and *PavWOX13B* had increased expressions upon drought stress. In addition, the transcription factor of *PavWOX4* and *PavWOX13A* was localized in the nucleus, confirming the estimated results. Our findings lay the foundation for determining the expression patterns and functions of the *PavWOX* gene family within sweet cherry and shed more light on the underlying regulatory mechanisms.

**Keywords:** sweet cherry; WOX; transcription factor; gene expression



## 1. Introduction

As a plant-specific transcription factor, the WUSCHEL-related homeobox (WOX) family contains one homeodomain (HD) involving 65 amino acid residues [1], and it is critical for embryogenesis, plant cell division, organ formation, and stem cell stability [1,2]. These functions are closely related to their ability to promote cell division and prevent premature differentiation of immature cells. *WOX* genes have been detected within diverse species, such as *Arabidopsis thaliana* [1], *Medicago truncatula* [3], *Oryza sativa* [4], *Brassica napus* [5], *Triticum aestivum* L. [6], *Zea mays* [7], *Citrus sinensis* [8], *Populus trichocarpa* [9], *Picea abies* [10] and *Pinus pinaster* [11], with family member numbers of 15, 11, 12, 52, 43, 12, 11, 18, 10, and 14, respectively. *WOX* gene family members are greatly different in different species, which may be due to gene duplication events during the long-term evolutionary process. It is speculated that there are also differences in gene function. From the perspective of evolutionary relationships, the WOX proteins of *Arabidopsis* can be divided into three branches: the WUS clade (WUS, WOX1–7), the intermediate clade (WOX8, 9, 11–12), and the ancient clade (WOX10, 13–14) [12]. Current research shows that the genome of *Pinus pinaster* contains at least 14 members covering all major *WOX* gene family evolutionary branches, and gymnosperms contain one *WOX* gene, representing

the transition between the intermediate and WUS branching proteins, which have no homologous genes in angiosperms. Researchers first detected independent transcripts of *WUS* and *WOX5* in gymnosperms [11].

It has been reported that *WOX* genes are extensively related to root, leaf, flower, stem, fruit, seed, and embryo development, and they exert a significant effect on environmental stress responses (like cold, drought, and salt responses) in plants. *WUS* is crucial for primary root, lateral root, and plant type development [13–15]. *WOX1* and *WOX3* regulate the development and morphogenesis of leaves [16,17]. *WOX2*, *WOX8*, and *WOX9* are involved in embryonic development [18–21]. *AtWOX5* and *AtWOX7* are activated by *AtWOX11/12* and specifically expressed during the root primordium stage, affecting the rate of cell division and adventitious root regeneration of the root primordium. When the function of *WOX5/7* is absent, adventitious root primordial cell division becomes disordered, and apical differentiation becomes abnormal [22]. In poplars, the overexpression of *PtoWOX5a* elevates adventitious root numbers but reduces the adventitious root length [23]. *AtWOX14* is specifically expressed in early formed lateral roots (LRs) and developing anthers [24]. In *Arabidopsis thaliana*, *AtWOX13* and *AtWOX14* affect flowering. *AtWOX13* is expressed in the vascular system, stigma, and pistil of a flower age of 13/14 and is associated with the flowering transition. After flowering, the presence of the *wox14* mutant leads to severe stamen defects, being incomplete and shorter than the pistil and thus preventing effective fertilization and leading to ovule abortion [24]. In *Arabidopsis thaliana*, the *PRETTY FEW SEEDS2/WOX6* gene is related to seed development regulation. For most pfs2 ovules, the embryo sac was aborted or showed anatomical abnormalities during development [25]. *CsWOX9* is expressed in developing cucumber fruits (*Cucumis sativus* L.), but less in the shoot apex and axillary buds. *CsWOX9* overexpression within *Arabidopsis thaliana* results in elevated branching and rosette leaves and shorter siliques in transgenic plants [26]. *GhWOX4* regulates drought stress in cotton (*Gossypium hirsutum*) by controlling vascular system growth. The knockout of *GhWOX4* leads to a decrease in the stem width, severe vascular growth impairment, and significantly reduced drought resistance in transgenic cotton. Conversely, its ectopic expression within *Arabidopsis* enhances drought stress resistance in plants. In addition, a GO enrichment analysis revealed that some transcription factors (TFs), such as MYB, AP2-ERF, MYC-bHLH, HB, bZIP, WRKY, HSF, GRAS, NAC, LOB, AUX/IAA, and C2C2-Dof, as well as plant hormones, may be critical for regulating plant development and drought resistance mediated by *GhWOX4* [27]. *OsWOX13* is associated with drought resistance in rice. *OsWOX13* overexpression driven by a rab21 promoter leads to enhanced drought resistance while causing flowering to occur 7–10 days earlier. A further analysis revealed that *OsWOX13* can activate the drought response genes *OsDREB1A* and *OsDREB1F* by binding to the ATTGATG motif in their promoters, thereby mediating rice's response to drought stress [28]. *PagWOX11/12a* enhances plant drought resistance in poplar by promoting root elongation and biomass growth and modulating gene levels associated with scavenging reactive oxygen species [29,30]. Based on these results, *WOX* genes are crucial for drought stress responses. However, current research on the effect of *WOX* on regulating plant drought resistance is still very limited, and related studies are mainly focused on plant species like *Arabidopsis*, rice, poplar, and cotton. There is very little research on drought stress response regulation via *WOX* within other plants, especially fruit trees. In addition, there is no report on the identification and analysis of the *WOX* gene family in sweet cherry.

In order to study the role of the *WOX* gene in the development and drought resistance of sweet cherry, this work detected eight *WOX* genes in the *Prunus avium* L. genome and later characterized the corresponding structure and protein sequence profiles. Thereafter, the *WOX* expression patterns in five tissues were determined, as well as drought stress treatment. As a result, *WOX* members within *Prunus avium* L. and the potential relation between *PavWOX13A* and drought stress were identified. Our findings shed more light on the effect of *WOX* on the modulation of drought stress within woody plants.

## 2. Materials and Methods

### 2.1. Plant Materials and Culture Conditions

Sweet cherry trees were obtained from a sweet cherry plantation in Tianshui City, Gansu Province, China (105°65′56″ E, 34°53′27″ N). Samples of buds, stem, leaves, and flowers were collected from 8-year-old cherry trees. Root samples were collected from 1-year-old seedlings. All samples were immediately frozen in liquid nitrogen and stored at $-80\,^\circ$C.

To check the subcellular localization and genetic transformation, we used tobacco seedlings. To disinfect the surfaces of *Nicotiana benthamiana* seeds, they were put into 2.0 mL centrifuge tubes and immersed in a 70% ethanol and 10% $H_2O_2$ solution (1 mL each) for 30 s and 15 min, respectively. Thereafter, seeds were washed with distilled water 5–6 times prior to being transferred into sterile Petri dishes that contained 4-layer moist sterile filter paper. After adding sterile water (2 mL), we incubated the dishes in the dark at 28 °C for a 5-day period. Seeds were subjected to inoculation on MS solid medium following germination and cultivated during the photoperiod for tobacco transients. Culture conditions: temperature of 21–23 °C, light/dark cycle of 16 h/8 h.

### 2.2. WOX Gene Identification in Sweet Cherry and Bioinformatics Analysis

To identify the *WOX* gene family within *Prunus avium*, *AtWOXs* were acquired based on the *Arabidopsis* Information Resource (TAIR) database (https://www.arabidopsis.org/) (accessed on 3 June 2022). We utilized the homeobox domain amino acid sequence as a BLASTP query in *Prunus avium* Tieton Genome v2.0 (https://www.rosaceae.org/Analysis/9262820) (accessed on 6 June 2022) with an e-value of $10^{-5}$. Then, the *Prunus avium* potential WOX protein sequence domain was analyzed with the hidden Markov model (HMM) (PF00046). Later, the domain analysis programs SMART (http://smart.embl-heidelberg.de/) (accessed on 8 June 2022) [31] and Pfam (http://pfam.xfam.org/) (accessed on 8 June 2022) [32] were employed for a manual examination of protein sequences with default parameters. The Compute pI/Mw approach on the ExPASy website (http://web.expasy.org/compute_pi) (accessed on 5 June 2022)was utilized to predict the pI and molecular weight. Protein–protein subcellular localization was predicted with Plant-mPLoc (http://www.csbio.sjtu.edu.cn/bioinf/plant-multi/) (accessed on 6 June 2022).

### 2.3. Phylogenetic Analyses

For *Arabidopsis thaliana*, *Picea abies*, *Populus trichocarpa*, and *Prunus persica*, WOX family protein sequences were acquired based on the NCBI protein database (https://www.ncbi.nlm.nih.gov/protein/) (accessed on 13 June 2022) and Plant Transcription Factor Database v5.0 (http://planttfdb.gao-lab.org/index.php) (accessed on 3 June 2022). Full-length protein sequences were subjected to multiple sequence alignment using Muscle. The neighbor-joining (NJ) approach was utilized to build a phylogenetic tree of full-length sequences through 1000 bootstraps in the MEGA 7.0 software [33].

### 2.4. Analyses of Gene Structures, Conserved Motifs, and Chromosomal Locations

Gene Structure Display Server 2.0 (GSDS, http://gsds.gao-lab.org//) (accessed on 12 June 2022) was used to identify exon/intron structures of *PavWOX* genes [34]. Protein-conserved motifs were analyzed using the Multiple Em for Motif Elucidation (MEME) program (https://meme-suite.org/meme/tools/meme) (accessed on 15 June 2022) with the following parameters: a repetition number of 0 or 1; a constrained optimal motif width of 30–70 residues; and a maximal motif number of 10. The SMART database was used to annotate MEME motifs. We later obtained chromosomal locations according to *Prunus avium* Tieton Genome v2.0-derived genome data [35]. Later, the MG2C database (Map Gene2 Chromosome v2, http://mg2c.iask.in/mg2c_v2.0/) (accessed on 16 June 2022) was employed to map these data to the chromosomes.

### 2.5. Promoter Analysis

Using TBtools, we obtained the promoter sequences of different *PavWOX* genes (2 kb sequences upstream) from the GDR database. In addition, PlantCARE (http://bioinformatics.psb.ugent.be/webtools/plantcare/html/ (accessed on 6 December 2022) was employed to analyze cis-acting elements, while TBtools was applied for visualization [36,37].

### 2.6. Gene Expression Analysis

*PavWOX* gene expression data within diverse plant tissues were originally downloaded based on transcription data from the EMBL-EBI database (codes SUB7211514) (https://www.ebi.ac.uk/, accessed on 10 December 2022), including sweet cherry buds, leaves, stems, flowers, and fruits. Drought treatment data are available in the NCBI Sequence Read Archive (SRA) database (accession numbers SRP095080 (*Prunus mahaleb* CDR-1) and PRJNA704726 (*P. cerasus* × *P. canescens* Gisela 5)). The Hisat2 v2.1.0 program [38] was used to map raw transcriptome data in the *P. avium* genome, while the Counts v2.0.0 feature [39] and the Python script were employed to process transcripts per million (TPM) for every *PavWOX*. At last, OmicStudio tools were employed to construct a heatmap showing *PavWOX* genes (https://www.omicstudio.cn/tool) (accessed on 3 July 2023).

### 2.7. RNA Isolation, cDNA Synthesis, and Gene Cloning

The total RNAs were extracted from the buds, stem, leaves, roots, and flowers of sweet cherry with a MiniBEST Plant RNA Extraction Kit (TaKaRa, Beijing, China). For the extracted RNA, its quality and purity were quantified using a NanoDrop8000 (Thermo-Scientific, Massachusetts, USA) through 1.2% agarose gel electrophoresis. Afterwards, a PrimeScript™ RT reagent kit (TaKaRa, Beijing China) was used to prepare first-strand cDNA from total RNA (2 μg) in line with specific protocols and then diluted for PCR amplification. Primer3 (v.0.4.0) software (http://bioinfo.ut.ee/primer3-0.4.0/primer3/input.htm) (accessed on 16 March 2024) was used to prepare specific PCR primers (Table S1). The amplified products were checked via agarose gel electrophoresis.

### 2.8. Expression Analysis of PavWOX by qRT-PCR

The total RNA was used to synthesize cDNA with the FastKing RT Kit (KR116 (Tian Gen Biotech, Beijing, China). qRT-PCR was performed using the Realtime PCR Super mix (Vazyme Biotechnology, Nanjing, China) with an Applied Biosystems 7500 real-time PCR system. The Cyclophilin 2 (CYP2: TC1916) gene was used as an internal control. Three biological replicates were used for an expression analysis. The primers are listed in Table S1.

### 2.9. Plasmid Construction and Subcellular Localization of PavWOX13A

We amplified the *PavWOX13A* coding sequence within cDNA in sweet cherry and inserted it into the plant expression vector pCAMBIA1304 to produce *35S::PavWOXs-GFP* constructs using an EasyGeno Assembly Cloning kit (Tiangen, China), with pCAMBIA1304-GFP as the positive reference. The plasmid of the *35S::PavWOX13A-GFP* construct was introduced into *A. tumefaciens* GV3101 by freeze–thawing. Then, we immediately froze bacteria liquid within liquid nitrogen and preserved it at −80 °C after mixing it with 40% glycerol (1:1) [9]. The leaves of 1-month-old tobacco were utilized for the transient expression of PavWOX13A-GFP fusion proteins. An Ultra-VIEW VoX 3D Live Cell Imaging System Spinning Disk confocal laser scanning microscope (PerkinElmer, Waltham, MA, USA) was employed to observe green fluorescence after being infected for 3 days.

### 2.10. Data Analysis Method

In this experiment, Microsoft Excel 2010 and SPASS 20.0 software were used for data processing and statistical analysis. The error bars indicate the Standard Deviation (SD) from three biological replicates. A one-way ANOVA was used for statistical analysis, and

the LSD method was used to compare the measured data. The asterisks indicate significant differences; * $p < 0.05$, ** $p < 0.01$.

## 3. Results

### 3.1. WOX Gene Family Identification within Sweet Cherry

Using the WOX protein sequence in *Arabidopsis* as a reference, conserved domains were homologously aligned and identified for screening and identifying *WOX* gene family members. Altogether, there were eight *WOX* genes obtained from sweet cherry. According to the homology with *Arabidopsis thaliana WOX* genes, they were named *PavWUS*, *PavWOX1*, *PavWOX2*, *PavWOX4*, *PavWOX5*, *PavWOX9*, *PavWOX13A*, and *PavWOX13B*. The chromosomal locations of these genes were then determined (Figure 1). As a result, the *PavWOX* genes exhibited an uneven distribution in sweet cherry chromosomes. Two *PavWOX* genes (*PavWUS* and *PavWOX13B*) are located on chromosome 7, and two *PavWOX* genes (*PavWOX1* and *PavWOX13A*) are located on chromosome 5. Just one individual *PavWOX* gene was located on each of the remaining four chromosomes (chromosomes 1, 2, 4, and 6). As shown in Table 1, the eight PavWOXs had molecular weights in the range of 20439.96 Da (PavWOX5)–45466.65 Da (PavWOX1), lengths in the range of 180 (PavWOX5) AAs–413 (PavWOX9) AAs, and pI values in the range of 5.32 (PavWOX13A)–9.30 (PavWOX4 and PavWOX5). A subcellular localization prediction analysis found that every PavWOX protein was localized in the nucleus.

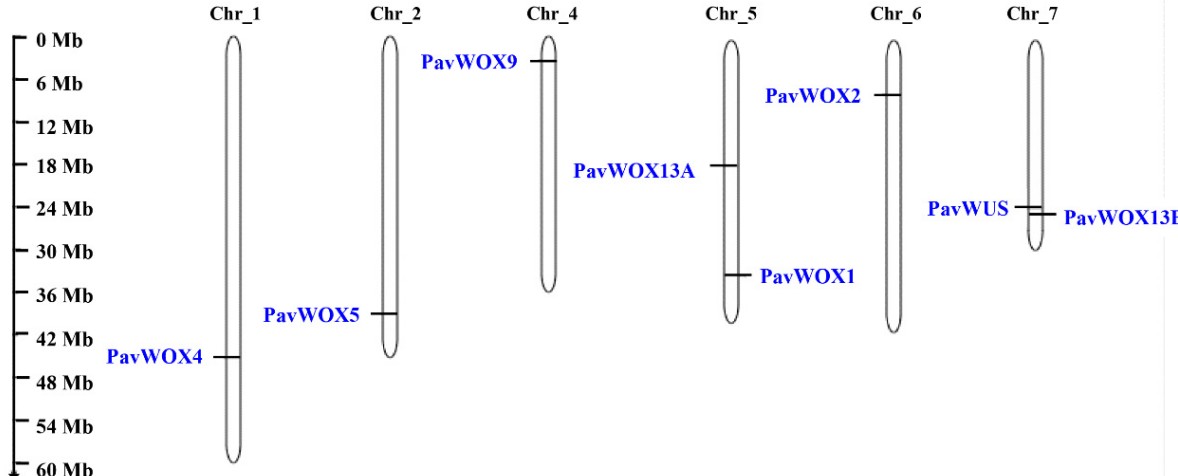

**Figure 1.** The chromosomal localizations of these eight *WOX* genes in sweet cherry. The chromosome number is presented on top of the chromosomes. The scale on the left is in megabases (Mb).

**Table 1.** Information of *WOX* gene family in sweet cherry.

| Gene Name | Gene ID | Genomic Position | Protein Length (aa) | Molecular Weight (Da) | Isoelectric Point | Subcellular Localization |
|---|---|---|---|---|---|---|
| *PavWUS* | FUN_038736 | chr_7: 24441104–24442540 | 305 | 33,845.92 | 5.91 | Nucleus. |
| *PavWOX1* | FUN_026584 | chr_5: 34401315–34403970 | 403 | 45,466.65 | 8.49 | Nucleus. |
| *PavWOX2* | FUN_019165 | chr_6: 7919462–7920784 | 261 | 29,121.23 | 7.71 | Nucleus. |
| *PavWOX4* | FUN_005951 | chr_1: 47077176–47078228 | 224 | 25,281.56 | 9.30 | Nucleus. |
| *PavWOX5* | FUN_012296 | chr_2: 40510721–40511406 | 180 | 20,439.96 | 9.30 | Nucleus. |
| *PavWOX9* | FUN_032123 | chr_4: 3475851–3477812 | 413 | 45,418.52 | 6.87 | Nucleus. |
| *PavWOX13A* | FUN_024119 | chr_5: 18202603–18205376 | 220 | 25,412.49 | 5.32 | Nucleus. |
| *PavWOX13B* | FUN_038933 | chr_7: 25450872–25453321 | 273 | 31,202.77 | 5.59 | Nucleus. |

### 3.2. Phylogenetic Analyses of the WOX Gene Family

Through the construction of an unrooted phylogenetic tree, we examined the evolutionary and phylogenetic relationships of 15 *Arabidopsis* WOX proteins with 8 PavWOXs. WOX proteins of sweet cherry were further divided into the ancient clade and the intermediate clade, with the WUS clade having a maximum of five genes and the intermediate clade having a minimum of one gene (Figure 2). Based on phylogenetic tree clustering, *PavWUS, PavWOX1, PavWOX2, PavWOX4*, and *PavWOX5* all belonged to the WUS clade, and *PavWOX13A* and *PavWOX13B* were in the ancient clade, whereas *PavWOX9* was in the intermediate clade.

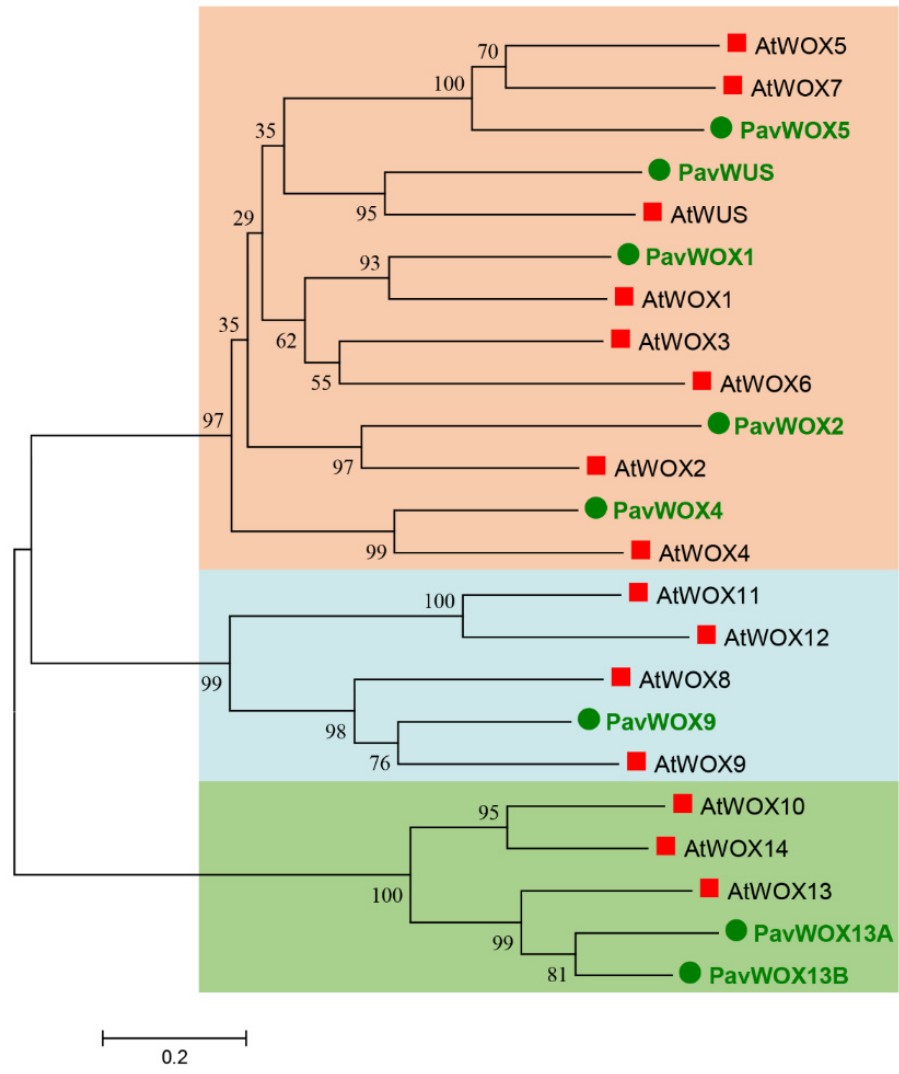

**Figure 2.** A phylogenetic tree showing WOX family proteins in *Arabidopsis thaliana* and sweet cherry. The numbers in all nodes from this phylogenetic tree represent bootstraps.

To better understand the phylogenetic relationships among *WOX* gene family members, we utilized full-length WOX protein amino acid sequences from *Arabidopsis thaliana*, *Prunus persica*, *Populus trichocarpa*, *Picea abies*, and sweet cherry to construct a phylogenetic tree (Figure 3). As suggested by this phylogenetic tree, *PavWOX* genes have a close genetic relationship with *P. persica* of the Rosaceae family, suggesting the conserved evolution of plant *WOXs*. Compared to peach, *WOX3* and *WOX11* did not have any homologous genes in the sweet cherry genome. There are two *WOX13* paralogue genes in peach and sweet cherry, which may have new functions (Figure 3).

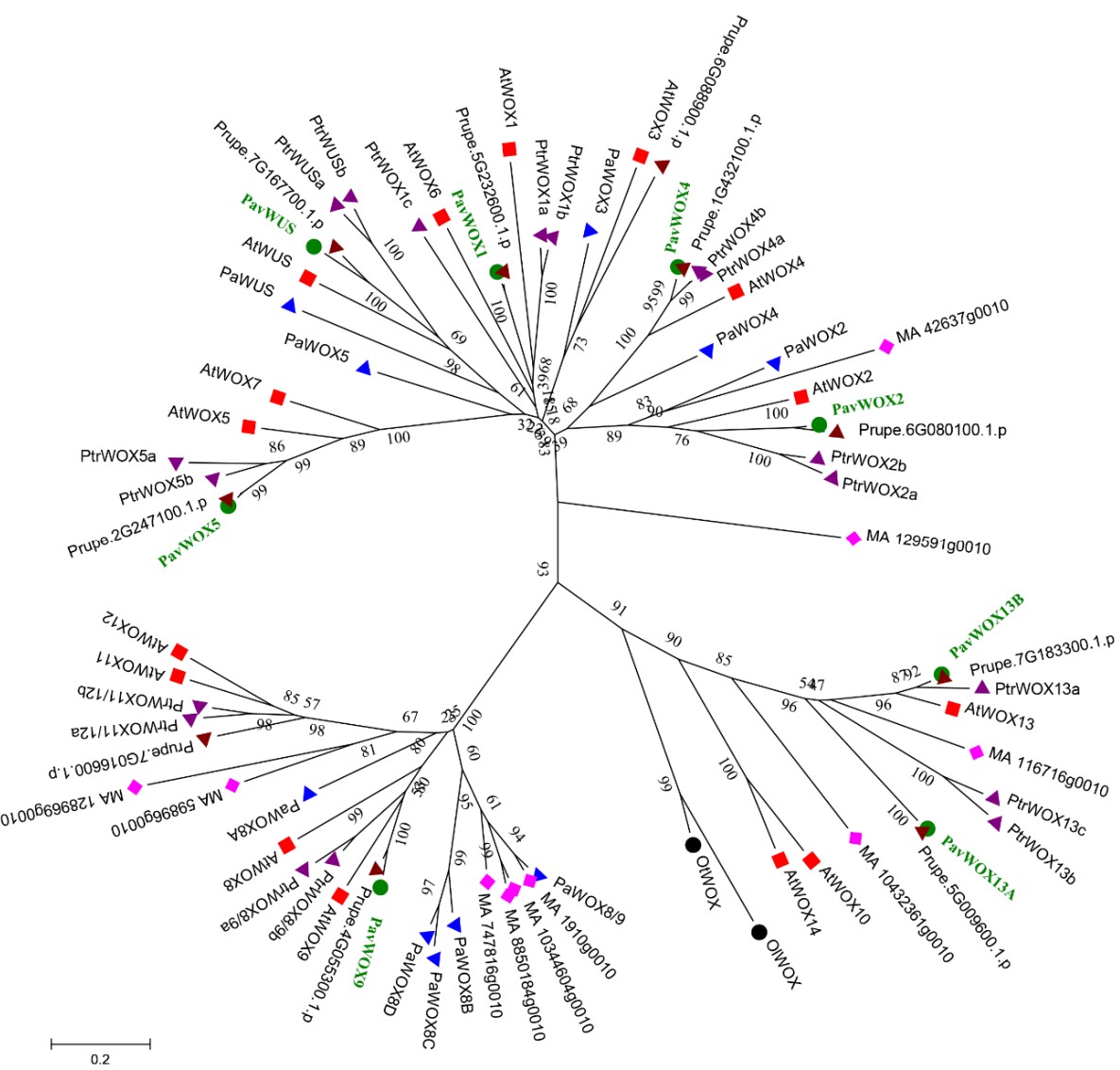

**Figure 3.** A phylogenetic tree showing WOX proteins from *Arabidopsis thaliana*, *Prunus persica*, *Populus trichocarpa*, *Picea abies*, and sweet cherry. Table S2 displays accession numbers. The data on the branches indicate the reliability percentages of the bootstraps according to 1000 replications. OlWOX and OtWOX were used as outgroups.

### 3.3. WOX Gene Structures and Conserved Motifs within Sweet Cherry

As for sweet cherry *WOX* genes, their structures and conserved motifs were examined to determine the structures. Additionally, to analyze the relations between gene structures, conserved motifs, and evolution, we also built an NJ phylogenetic tree similar to the findings in Figure 2 (Figure 4). The gene structures were analyzed; *PavWOXs* had 2–4 exons, and members in one sub-clade exhibited identical or close intron/exon patterns (Figure 4).

We also detected conserved motifs in eight PavWOX proteins with the MEME Software (Version 5.5.5). Ten motifs were identified in the *PavWOX* gene family (Figure 4, Table S3). All PavWOX proteins had motifs 1 and 2, suggesting that there were conserved gene motifs in a specific category in the evolution process. Additionally, motif 4 was present in all members of the modern branch. In contrast, motif 8 is only shared by PavWUS and PavWOX2, motif 9 is only shared by PavWOX1 and PavWOX9, and motif 10 is only shared by PavWOX2 and PavWOX13B. PavWOX13A and PavWOX13B also had motifs 3, 5, and 6. Genes in one clade showed similar motif locations and distributions (Figure 4).

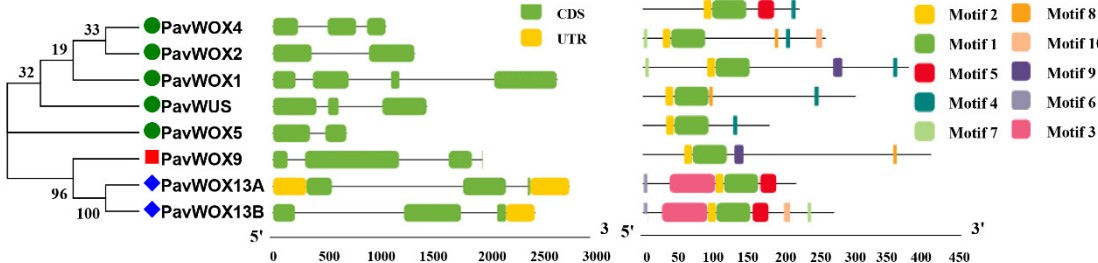

**Figure 4.** Gene structures of sweet cherry *WOXs* analyzed based on phylogenetic relationships. Exon–intron structures were analyzed based on GSDS database. Yellow boxes, green boxes, and black lines stand for upstream/downstream, exons, and introns, respectively. Conserved motifs in sweet cherry *WOXs* are based on phylogenetic relationship. Every motif was detected through MEME database by using complete amino acid sequences in sweet cherry *WOXs*.

### 3.4. Multiple Sequence Alignment Analysis on WOX Proteins within Sweet Cherry

According to the multi-sequence alignment analysis of *Arabidopsis thaliana* and sweet cherry WOX proteins via DNAman, each protein possessed one conserved HD domain (Figure 5). Some residues are composed of homeobox domain motifs containing three helixes separated by a loop and a turn (Figure 5). The WUS-box domain was shared by five members from the modern/WUS clade (Figure 5).

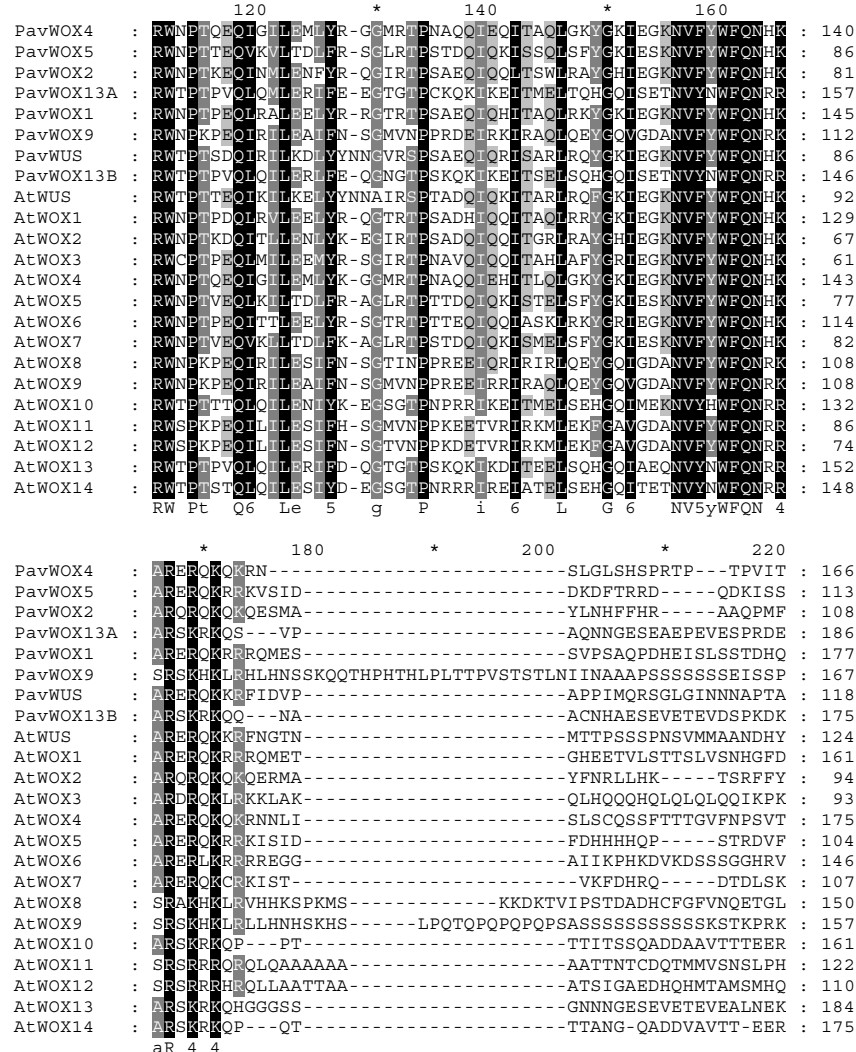

**Figure 5.** The protein sequence alignment of the conserved domain of the WOX family in sweet cherry.

### 3.5. Promoter Component Analysis of WOX Genes in Sweet Cherry

To predict the transcription features and functions of *PavWOX* genes, we estimated *cis*-regulatory elements via PlantCARE using a 2 kb promoter in each gene. These promoters included various *cis*-acting elements (Figure 6). Briefly, we discovered elements related to hormones, stress, and development. Specifically, hormone-responsive elements were ABA-responsive elements (ABREs), MeJA-responsive elements (CGTCA motif-containing elements), gibberellin (GA)-responsive elements (P-boxes, GARE motif-containing elements, and TATC-boxes), salicylic acid-responsive elements (TCA-elements), and auxin-responsive elements (TGA-elements, AuxREs, and AuxRR-core elements). We also predicted abiotic stress-responsive elements using regulatory anaerobic inductor elements (AREs), drought-responsive elements binding to MYBs (MBSs), anoxic-specific induction-responsive elements, defense- and stress-responsive elements, and low-temperature-responsive elements (LTRs). Additionally, we predicted O2-sites and CAT-box elements separately from development-related *cis*-acting elements. Frequently observed *cis*-acting elements within *PavWOX* promoters included ABREs (ABA-related), AREs (anaerobic induction), and CGTCA motif-containing elements (MeJA-related). Therefore, *PavWOXs* are related to plant development and stress response.

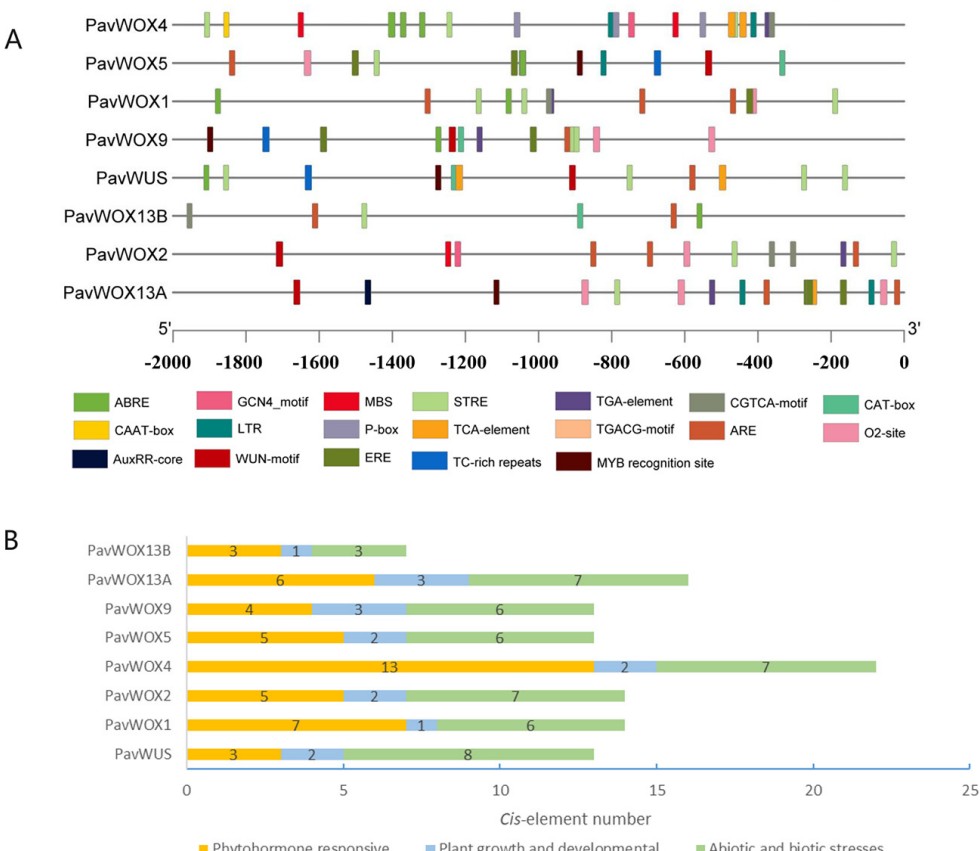

**Figure 6.** The *cis*−element analysis in *PavWOX* promoters. (**A**) The schematic was constructed according to the *PavWOXs* promoter sequences analyzed through PLACE. Each color indicates one element at diverse positions. (**B**) The amount of *cis*−elements in three groups for each *PavWOX* promoter.

### 3.6. Differential WOX Gene Expression within Sweet Cherry

To understand the expression characteristics of the eight *PavWOXs* within sweet cherry tissues (including dormancy/flower buds, young/mature leaves, first bloom, fruits, and stems) in different developmental periods, RNA data in public databases were analyzed. The results showed that the *WOX* gene levels within sweet cherry were markedly different, suggesting that they may have different biological functions (Figure 7). In addition,

transcript abundances of *PavWOX9*, *PavWOX13A*, and *PavWOX13B* could be detected in every tested tissue. On the contrary, *PavWOX2* and *PavWOX5* are poorly expressed in these tissues and are specifically expressed in the buds and young leaves, respectively. *PavWOX4*, besides not being expressed in fruits, is stably expressed in various tissues, but the overall level is not high. The expression of *PavWUS* is the highest in flower buds and low or undetectable within additional tissue types (Figure 7). Consequently, *WOX* genes are specifically related to regulating sweet cherry tissue growth.

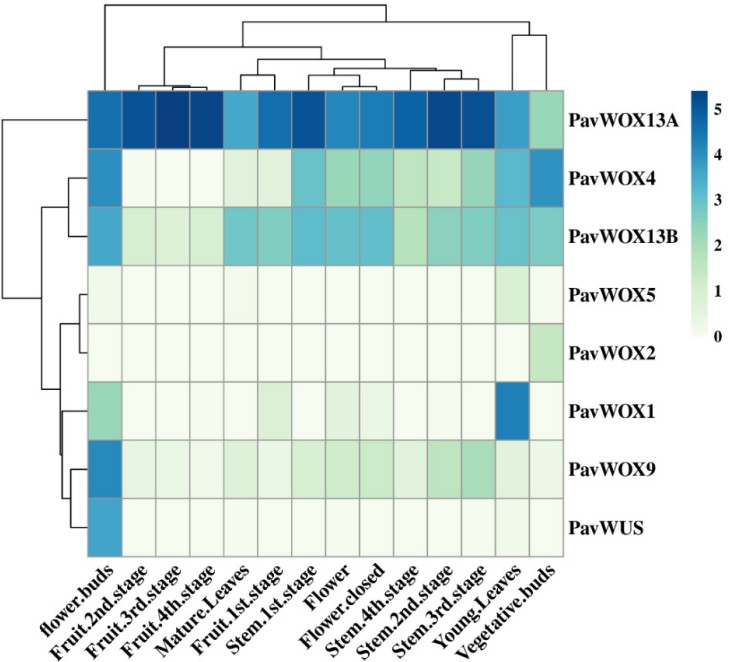

**Figure 7.** Sweet cherry WOXs' expression patterns within diverse tissues of dormant/flower buds and fruit/stems in four developmental periods, young/mature leaves, first blossom, and flowers. Heatmap was constructed according to log2-based fold changes denoted in color as the scale.

To investigate the potential functions of *PavWOXs* in developmental processes, the transcriptional profiles of eight PavWOX genes in the buds, roots, stems, leaves, and flowers were studied by qRT-PCR (Figure 8). *PavWUS*, *PavWOX1*, and *PavWOX9* are mainly expressed in the leaves. It is worth noting that *PavWOX4*, *PavWOX5*, *PavWOX13A*, and *PavWOX13B* are strongly expressed in the roots. In addition, *PavWOX2* is mainly expressed in the flowers.

### 3.7. PavWOX Expression under Drought Stress

To understand the levels of the eight *PavWOX* genes in sweet cherry upon drought stress, we analyzed their expression based on RNA data obtained from public databases. The *PavWOX* genes had very different responses to drought stress. Under drought treatment, *PavWOX13A* and *PavWOX13B* expressions increased in the roots compared with the leaves, and they also increased in CDR-1, a rootstock with strong drought resistance, and in Gisela 5, a rootstock with weak drought resistance. *PavWOX4* and *PavWOX5* were mainly expressed in the roots, and CDR-1 expression increased relative to Gisela 5 rootstocks. Consequently, *PavWOX* genes are probably related to drought stress responses in sweet cherry (Figure 9).

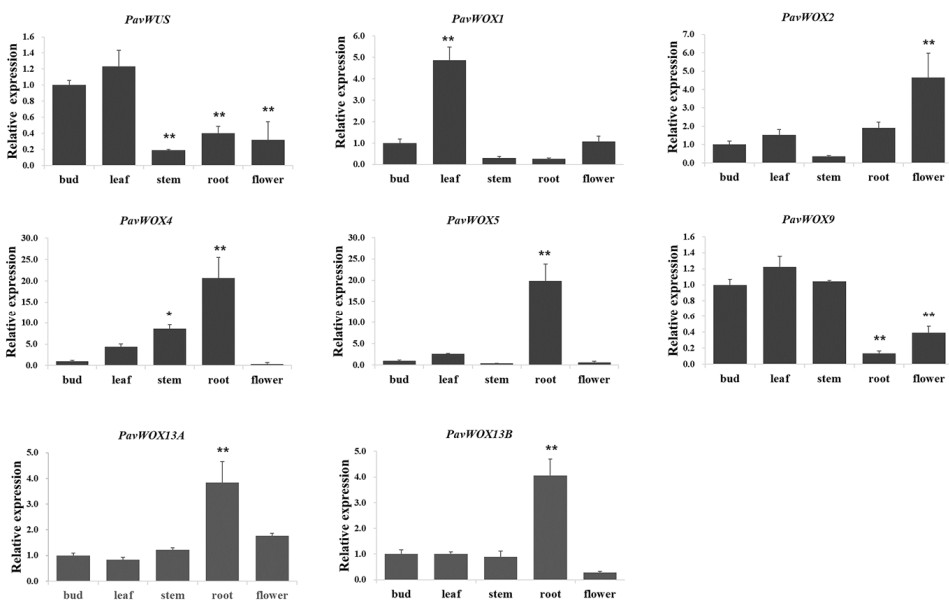

**Figure 8.** *PavWOX* expression in roots, leaves, stems, buds, and flowers of sweet cherry seedlings as detected by qRTPCR using sweet cherry CYP2 as an endogenous control. Bars show SD from three biological replicates. Error bars indicate Standard Deviation (SD) from three biological replicates. Asterisks indicate significant differences; * $p < 0.05$, ** $p < 0.01$.

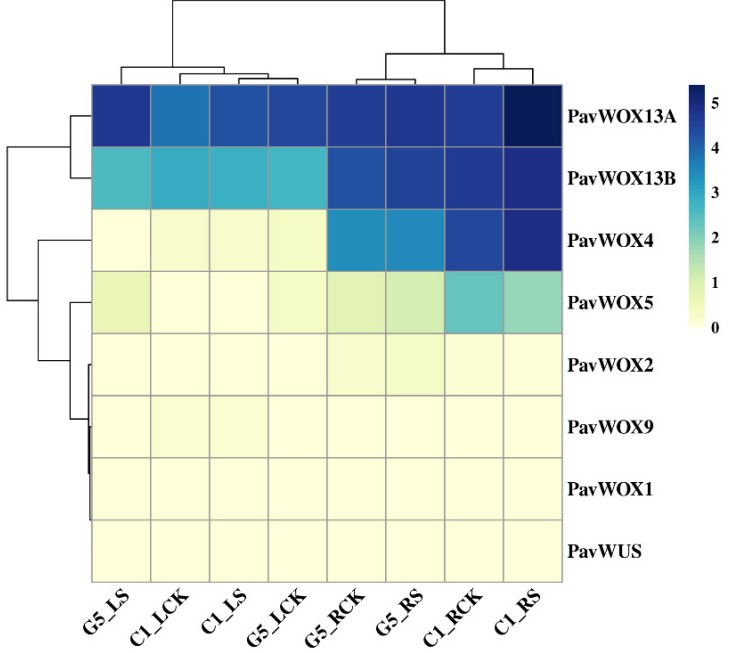

**Figure 9.** Expression level analysis of *PavWOX* under drought treatment. C1 and G5 represent CDR-1 and Gisela 5 rootstocks, respectively. LS and LCK indicate leaf treatment group and leaf control group, respectively, and RS and RCK indicate root treatment group and control group, respectively.

### 3.8. Nuclear Localization of PavWOX4 and PavWOX13A

The subcellular localization of WOX proteins in sweet cherry was predicted in the nucleus. To verify the subcellular localization of PavWOX4 and PavWOX13A, double enzyme digestion was conducted to prepare pEGOEP35S-PavWOX4-GFP and pE-GOEP35S-PavWOX13A-GFP fusion expression vectors. Then, the recombinant plasmid with the correct sequence was transfected into *Agrobacterium* GV3101 competent cells. After tobacco transient expression, a confocal laser scanning microscope was employed to observe the

fluorescence signals. As shown in Figure 9, the PavWOX4 and PavWOX13A proteins were localized in the nucleus (Figure 10A,B), whereas the positive control exhibited expression within each organelle (Figure 10C).

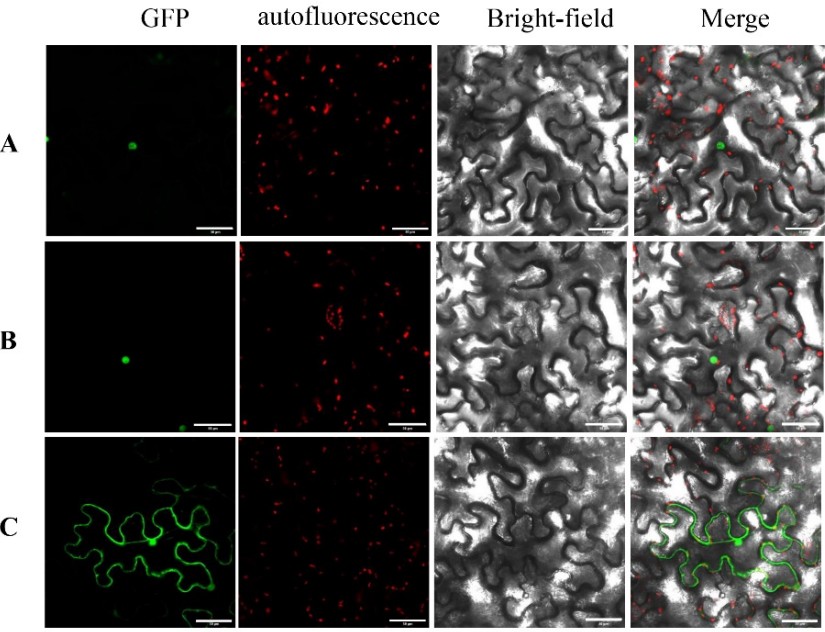

**Figure 10.** Subcellular localization of PavWOX4 and PavWOX13A proteins within tobacco leaf cells and onion epidermal cells with transient expression of GFP-PavWOX4 and GFP-PavWOX13A fusion proteins. Images showing tobacco leaf cells subjected to agroinfiltration with (**A**) GFP-PavWOX4 fusion protein, (**B**) GFP-PavWOX13A fusion protein, and (**C**) GFP alone. Scale bar: 50 μm.

## 4. Discussion

*WOXs* exhibit plant specificity and are greatly related to plant development and stress processes. Numerous plant genome sequences have been published, with *WOX* genes being detected in some plants. In this work, eight *WOX* genes were detected altogether in the *Prunus avium* genome, showing uneven distributions in six chromosomes (Figure 1). In five Rosaceae species, the genome size was not directly related to the *WOX* gene family member number, including *Prunus avium* (8 *WOXs*, 344.29 MB), *Pyrus bretschneideri* (9 *WOXs*, 271.9 MB), *Fragaria Vesca* (14 *WOXs*, 240 MB), *Prunus persica* (10 *WOXs*, 224.6 MB), and *Prunus mume* (10 *WOXs*, 201 MB) [40]. This was not consistent with prior studies on *Populus trichocarpa* [9] and *Malus domestica* [41], which discovered the contribution of recent genome-wide duplication events to *WOX* gene family number expansion. The *WOX* gene family is classified into three clades. In *Arabidopsis*, the ancient clade includes WOX10, WOX13, and WOX14 proteins. Two ancient WOX proteins that were highly homologous in the sequences were obtained from sweet cherry: PavWOX13A and PavWOX13B (Figure 2). Poplar, apple, and strawberry contain three ancient WOX proteins: PtrWOX13a, PtrWOX13b, and PtrWOX13c [9]; MdWOX13a, MdWOX13b, and MdWOX13c [41]; and FvWOX13A, FvWOX13B, and FvWOX13C [40], respectively. In contrast to *Arabidopsis*, no homologous proteins for WOX3 and WOX6 were found in the sweet cherry genome in the WUS/modern clade, and only WOX9 was found in the intermediate clade (Figures 2 and 3). This may result in the sweet cherry WOX protein exhibiting different functions.

Gene structural diversity has been previously identified as the key to multi-gene family evolution. The results of a *PavWOX* gene family structural analysis (Figure 5) were broadly the same as those from the phylogenetic analysis. In five Rosaceae species, different members of *WOXs* have one to five exons, and in *Picea abies* and *Phoebe bournei*, different members of *WOXs* have one to eight exons. Therefore, *WOX* gene functional diversity is probably related to a gain or loss of exons in *WOX* gene family evolution. Conserved

protein motifs have a critical effect on evolution. Members in one clade share similar motif locations and distributions (Figure 4).

Specific gene expression within diverse tissues can partially indicate functions in different tissues. The expression profiles of the *WOX* gene family were analyzed from numerous species. We found that the *WUS* gene in sweet cherry is expressed mainly in flower buds, which differs from the rest of the genes, which are expressed in the roots, leaves, stems, and SAMs [9,13,41]. Consequently, *PavWUS* may be important for maintaining flower bud differentiation. In *Arabidopsis*, *WOX1* regulates lateral growth and the shape of leaves [16,17]. *PavWOX1* is highly expressed within young leaves and also in flower buds as well as flowers. It probably has an important effect on flower buds and flowers. *WOX5* exerts a critical effect on adventitious root growth in *Arabidopsis thaliana* and poplar. Here, *PavWOX5* was specifically expressed in the roots (Figures 8 and 9). Therefore, it probably exerts a similar effect on *Prunus avium* to its homologs on *Arabidopsis* and poplar. In *Arabidopsis* and rice, *WOX4* promotes the differentiation of cambium or primary roots; in apples, *WOX4* induces adventitious root formation [41]. *PavWOX4* was stably expressed in the dormant/flower buds and stems in four developmental periods: young/mature leaves, first blossom, flowers, and root tissues. *AtWOX13* and *AtWOX14* were expressed in the roots and flowers. In contrast, *PavWOX13A* and *PavWOX13B* were universally expressed within nearly every tested tissue in sweet cherry (Figure 7) but had the highest expression in the roots (Figure 8). Similarly, it has been reported that *MdWOX13a*, *MdWOX13b*, and *MdWOX13c* can be detected in nearly every tested tissue [41]. Interestingly, *PavWOX* genes all contain plant hormone-responsive elements, including ABA-, MeJA-, gibberellin (GA)-, salicylic acid-, and auxin-responsive elements. Consequently, *PavWOX* genes may be involved in the regulation of sweet cherry growth and development.

*WOX* family genes are important for drought stress responses in plants. For instance, *GhWOX4* positively regulates drought tolerance in cotton [27]. The overexpression of *OsWOX13* leads to enhanced drought tolerance in rice [28]. *PagWOX11/12a* enhances plant drought resistance in poplar by promoting root elongation and biomass growth [29,30]. To explore the expression patterns of the eight *WOX* genes within sweet cherry under drought stress, the RNA-seq data of roots and leaves under drought stress treatments in sweet cherry were analyzed. As discovered, *PavWOX4*, *PavWOX5*, *PavWOX13A*, and *PavWOX13B* expressions increased under drought treatment. *PavWOX4*, *PavWOX5*, *PavWOX13A*, and *PavWOX13B* expressions increased in drought-resistant rootstock CDR-1 (Figure 8). Moreover, these genes include drought-responsive elements that could bind MYBs (MBSs), suggesting that they may be involved in drought stress responses. Current research on the effect of *WOX* on plant stress tolerance regulation is lacking. In this study, the *WOX* genes related to drought stress responses in sweet cherry were explored, which is significant for a deeper understanding of their molecular mechanisms in drought stress and provides new clues for breeding drought-resistant cherries.

## 5. Conclusions

In the present work, eight *PavWOX* genes were detected and classified into three clades. They were predicted to be localized on six chromosomes. The structures and conserved motifs of the members of this gene family were analyzed; genes in one clade exhibited similar structures, suggesting that their encoded proteins may have similar functions. Furthermore, the expression patterns and promoter *cis*-regulatory elements were analyzed, demonstrating that this gene family participates in development regulation and drought stress responses. In particular, *PavWOX5* may be involved in the regulation of root development. Our findings provide a further understanding of *WOX* gene functions within sweet cherry trees. However, the function of the WOX gene and its mechanism of drought resistance are not clear, which will be an important research direction in the future.

**Supplementary Materials:** The supporting information below is available at https://www.mdpi.com/article/10.3390/horticulturae10040370/s1, Table S1: The primers adopted in the present work; Table S2: The accession numbers of proteins in the present work; Table S3: Conserved motifs of WOX genes within sweet cherry.

**Author Contributions:** F.D. and H.W. were responsible for study conception and design, experiment implementation, data analysis, and manuscript drafting. F.D. and X.A. conducted sample collection, RNA extraction, and gene cloning. J.Y.U. contributed to English text editing in the present manuscript. F.D. and H.W. were in charge of project design coordination and manuscript writing. All authors have read and agreed to the published version of the manuscript.

**Funding:** The present work was supported by the scientific research project of Tianshui Normal University (no. CYZ2022-02) and the Natural Science Foundation of Gansu Province (no. 22JR11RE195).

**Data Availability Statement:** The data are contained within the article and Supplementary Materials.

**Acknowledgments:** We are grateful to An Feng, College of Horticulture, Northwest A&F University for helping with the data analysis.

**Conflicts of Interest:** The authors declare no conflicts of interest.

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
