# Peer review of "A Genome-Wide Analysis of the WUSCHEL-Related Homeobox Transcription Factor Family Reveals Its Differential Expression Patterns, Response to Drought Stress, and Localization in Sweet Cherry (Prunus avium L.)"

_horticulturae, doi:10.3390/horticulturae10040370_

Round 1
Reviewer 1 Report
Comments and Suggestions for Authors
The manuscript Genome-wide Analysis of WOX Transcription Factors Family Reveals Differential Expression Patterns and Localization in Sweet Cherry (Prunus avium L.) by Deng et al characterize the WOX transcription factors family in sweet cherry with different approaches. The function of different genes of this transcription family has been described in model plants and important crops. They are involved in many plant developmental processes and the number of genes varies within the different plant species. So, the characterization of this family in different plant species is relevant.
In general, the manuscript is well written, the results and the methods are clearly presented.
Minor revisions
P2 lines 44-46. This sentence is not clear.
P3 line 87. You are referred to plant species not plant varieties
Results
The two phylogenetic trees, figure 2 and 3, have almost the same information. Just figure 3 is needed.
It is necessary to discuss more about the relevance of the effect of cold in sweet cherry and the relation with the WUS family genes.
Author Response
Dear editor, thank you for reviewing our manuscript. We have completed the additions and revisions to the manuscript. Please see the attachment. thank you!

Reviewer 2 Report
Comments and Suggestions for Authors
This manuscript titled “Genome-wide Analysis of WOX Transcription Factors Family Reveals Differential Expression Patterns and Localization in Sweet Cherry (Prunus avium L.)” presents essential new data about the WOX transcription factors family in sweet cherry. There are several shortcomings that should be addressed.
Some major remarks:
Line No. 2: In the title: “Genome-wide Analysis of WOX Transcription Factors Family” should be “Genome-wide Analysis of the WOX Transcription Factors Family”
Line No. 159: Didn’t receive supplementary Table S1. Please include it in the revised file.
Line No. 265: Please divide the cis-regulatory elements into several groups, like hormone, developmental, and stress, and present how many cis-regulatory elements are present in each gene in a heatmap.
Line No. 268: Along with the public database, the author should perform their own sample to validate the expression in different organs.
Some minor remarks:
Line No. 68 and so on: Please maintain the same style for transgenic, like “CsWOX9” throughout the manuscript.
Line No. 98: Please modify the paragraph by introducing why you used tobacco seeds in the first line. (For checking the subcellular localization, we grow tobacco seeds by disinfecting with 70% ethanol and 10% H2O2 solution….)
Line No. 137: Add some references that analyze the 2kb promoter.
Line No. 249 and so on: Please write “cis-regulatory elements” instead of “cis- regulatory elements”
Author Response
Dear Reviewer, thank you for reviewing our manuscript. We have completed the additions and revisions to the manuscript. Please see the attachment. thank you!

Reviewer 3 Report
Comments and Suggestions for Authors
I have reviewed the manuscript titled "Genome-wide Analysis of WOX Transcription Factors Family Reveals Differential Expression Patterns and Localization in Sweet Cherry (Prunus avium L.)" by Fei Deng et al., which has been submitted to the journal Horticulturae. The objective of this study was to identify and characterize eight WOX genes in the Prunus avium L. genome, analyze their corresponding structures and protein sequences, investigate WOX expression patterns in five tissues, and assess their response to drought stress.
It's important to convey the specific focus of the study, especially if there are treatments involved. In this sense, the title: “Genome-wide Analysis of WOX Transcription Factors Family Reveals Differential Expression Patterns and Localization in Sweet Cherry (Prunus avium L.)”, lacks specificity regarding the conditions under which the differential expression patterns and localization are explored. In other words, the title doesn't provide information about the treatments or specific conditions that lead to the observed differential expression patterns. This omission could limit the reader's understanding of the experimental context and the significance of the findings.
The introduction mentions that current research on the effect of WOX genes on regulating plant drought resistance is limited, with a focus on specific plant varieties. To strengthen the introduction, consider emphasizing the existing gap in knowledge more explicitly. This could help set the stage for the significance of the current study within the context of addressing this gap. The introduction also concludes by introducing the current study on Prunus avium L. While the transition is clear, consider explicitly stating why investigating WOX genes in this specific fruit tree species is relevant or unique. This could be emphasized to underscore the importance of the forthcoming study.
Finally, it could be beneficial to explicitly state the objectives or aims of the current study at the end of the introduction. This would provide a clear roadmap for readers and help them understand the specific focus of the research.
The description of plant material and culture conditions is comprehensive, but including more details about the growth conditions, such as temperature, humidity, and light intensity during the cultivation period, could provide a clearer picture of the experimental setup.
In the section on promoter analysis, providing more information on the rationale behind selecting the 2 kb upstream sequences for promoter analysis and how this choice aligns with existing literature, or findings could enhance the credibility of the approach.
The gene expression analysis section describes the data sources and tools used but lacks information on the quantitative aspects of gene expression changes. Including some key quantitative results or statistical analyses would provide a more complete understanding of the gene expression patterns.
Statistical Analysis:
If statistical analyses were performed for any of the experimental data, it would be important to mention this explicitly. Including details on the statistical methods used and significance levels would strengthen the interpretation of the results.
Positive and Negative Controls:
In experimental sections involving molecular biology techniques, it's good practice to mention the use of positive and negative controls. This ensures that the experimental procedures are working as expected and helps validate the obtained results.
The conclusion could benefit from more contextualization of the findings within the broader scientific landscape. Consider mentioning how the study contributes to existing knowledge about WOX gene functions, especially in comparison to findings in other plant species. Additionally, including a brief section on potential future research directions could add depth to the conclusion. For example, what questions remain unanswered, and how might further investigations build upon the current study?
Author Response

(The authors gave the same response as above.)

Round 2
Reviewer 2 Report
Comments and Suggestions for Authors
I appreciate the author's diligent updating of the comments.
There are several things that should be considered:
Line 311: If it's possible, authors should change the Figure 8's color darker to improve visibility.
Line 317: The SRA number should be included in the publicly available RNASeq file used in the manuscript.
Author Response

(The authors gave the same response as above.)

Reviewer 3 Report
Comments and Suggestions for Authors
While it is true that several aspects have been improved, the authors must clearly address the changes made and respond to my questions. A question-and-answer form would be helpful. Maintaining effective communication throughout the peer review process is important. For example, regarding my concern:
"Statistical Analysis:
If statistical analyses were performed for any of the experimental data, it would be important to mention this explicitly. Including details on the statistical methods used and significance levels would strengthen the interpretation of the results.
Positive and Negative Controls:
In experimental sections involving molecular biology techniques, it's good practice to mention the use of positive and negative controls. This ensures that the experimental procedures are working as expected and helps validate the obtained results."
The authors have not responded at all.
They must respond to each point of my review, I insist.
Author Response

(The authors gave the same response as above.)

Round 3
Reviewer 3 Report
Comments and Suggestions for Authors
I have reviewed the modifications made, and I find that you have adequately responded to my observations. Therefore, I am satisfied with the current status of the paper.